# Compilation of Load Spectrum of PHEV Transmission Assembly and Its Simulation Application

**Baoqi Ma, Chongyang Han, Weibin Wu \*, Zhiheng Zeng, Chenyang Wan, Zefeng Zheng and Zhibiao Hu**

College of Engineering, South China Agricultural University, Guangzhou 510642, China;
mabaoqi@stu.scau.edu.cn (B.M.); 20202009003@stu.scau.edu.cn (C.H.); zengzhiheng@stu.scau.edu.cn (Z.Z.);
keith@stu.scau.edu.cn (C.W.); scauzzf@stu.scau.edu.cn (Z.Z.); huzhibiao1998@stu.scau.edu.cn (Z.H.)
\* Correspondence: wuweibin@scau.edu.cn; Tel.: +86-20-85-282-269

**Abstract:** This paper presents a method for compiling the load spectrum of the transmission assembly of plug-in hybrid electric vehicles (PHEVs). Based on the analysis of the control strategy of the test vehicle, the power flow transmission route in the transmission assembly is different under different operation modes, so it is necessary to divide different load spectrum blocks according to the operation mode. Based on the big data survey of China's national standard, it is determined that the typical working conditions are urban road working conditions, high-speed road working conditions, provincial road working conditions and poor road conditions. The mileage proportion of the various working conditions is 55:30:10:5, and the mileage of one cycle is 300 km. A total of three cycles are collected. After data processing and time-domain verification, based on the principle of maximum damage, the cycle with the largest pseudo damage is selected as the sample load data for load spectrum extrapolation. The rain flow counting method is used to count the sample load, and a two-dimensional kernel density estimation mathematical model with adaptive bandwidth is established to estimate the probability density function of the data. The extrapolated rain flow matrix is obtained through Monte Carlo simulation. The load spectrum of the two-dimensional rain flow matrix is transformed into a one-dimensional eight-stage program load spectrum by using a variable mean method, Goodman equation and equal damage principle theory. Finally, the fatigue life of the transmission assembly is simulated and calculated under the environment of Romax Designer simulation software. The two-dimensional kernel density estimation model with adaptive bandwidth is used to fit and extrapolate the load rain flow matrix of each hybrid mode of the PHEV, which solves the problem wherein the shape of the rain flow matrix of each hybrid mode of the hybrid electric vehicle is complex and difficult to fit. Finally, taking the after-sales maintenance data of this model from 2020 to the present as auxiliary proof, the failure components and the failure mileage life of the simulation test results are consistent with the results used by the actual users. This shows that the kernel density estimation model proposed in this paper can well fit the rain flow matrix of the PHEV load spectrum. The extrapolated load spectrum based on this model has high accuracy and authenticity. The method of compiling the load spectrum of the transmission assembly of a hybrid electric vehicle in this paper is effective.

**Keywords:** plug-in hybrid vehicle; transmission assembly; load spectrum; adaptive bandwidth kernel density estimation; simulation application

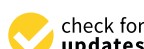



## 1. Introduction

Energy is the basis and lifeline of a country's economic development. The safety and supply of energy directly affects people's livelihood, safety and development within a country. At present, fossil energy is the main energy source in China, and carbon emissions also cause certain harm to the environment. As the environmental impacts of global warming become more and more important, 77 countries have committed to carbon neutralization by 2050. One of the major causes of global warming is road traffic carbon

emissions, which account for 18% of the total global carbon emissions. In 2020, China also set the goal of achieving a carbon peak by 2030 and carbon neutralization by 2060. Meanwhile, plans to ban fuel vehicles have begun in various parts of the world. In the face of an energy crisis and environmental pollution, it has become a consensus among global development to change the structure of energy use and develop and utilize new energy sources and related energy-saving technologies so as to reduce the dependence on nonrenewable energy sources and environmental pollution [1]. The development of new energy vehicles, driving and leading the energy industry revolution from the perspective of the transportation and automobile industry, is the main focus of current automotive technology research [2].

Internationally, as an important technical means of energy saving and emission reduction, new energy vehicles can reduce the emission of $CO_2$ and thus promote the international goal of carbon neutralization [3]. In recent years, the scale of the new energy automotive industry has been growing. New energy vehicles, including battery electric vehicles (BEVs) and plug-in hybrid electric vehicles (PHEVs), have become the second most popular power system in the European market [4], and the automotive market has become more and more active [5]. Compared with pure electric vehicles, plug-in hybrid electric vehicles do not have the problem of "mileage anxiety", while taking into account the advantages of pure electric vehicles when traveling short distances [6]. On the other hand, PHEVs still demonstrate the fuel-saving performance of traditional hybrid electric vehicles after the energy balance of the rechargeable energy storage system, which attracts wide attention from Chinese enterprises [7], including the national key laboratories of Hubei Province [8] and Chongqing [9], which are stronger in China. In 2020, China's PHEV production and sales around the world were 260,000 and 250,000, respectively.

As the main component of the PHEV powertrain, the transmission assembly plays the role of transmitting power flow and coupling power source power. Fatigue failure is a common failure form in the transmission assembly. In the actual driving process, it needs to bear the power output of the engine and drive motor. In the specific operating mode, it also needs to couple the power output of the engine and drive motor and transfer it to the drive axle. Once fatigue failure occurs, the whole vehicle will not run. Therefore, it is very important to predict the fatigue life of the transmission assembly in the process of automobile research and development. At present, the research on PHEVs mainly focuses on energy management, control strategies and structure optimization analysis, while there is little research on reliability analysis and there are few studies on the reliability of new energy vehicles based on load spectrum technology.

The technology of the load spectrum has been widely used in aerospace [10], rail transit [11], CNC machine tools [12] and automobiles [13]. The load spectrum is the data basis for the life design and reliability analysis of mechanical parts. The compilation of the load spectrum for a transmission assembly can improve the reliability of the fatigue test and is of great significance to improve transmission reliability. The main process of compiling the load spectrum is as follows [14]: determining typical test conditions, load measurement acquisition and data processing, statistical counting, load extrapolation and compiling program load spectrum according to certain methods and, finally, applying it to a reliability test or simulation. The compilation of the reliable load spectrum of a PHEV transmission assembly can improve the accuracy of the reliability test and promote the development of new energy vehicles.

Extrapolation of the load spectrum is the core step in compiling the load spectrum. Common extrapolation methods include time-domain extrapolation and rain flow matrix extrapolation. Time-domain extrapolation can retain the time sequence of loads and is suitable for loads with good stability. Wang [13] aimed at the periodic relationship between the transformation of cutting force and spindle speed in the cutting process of an NC machine tool; a BMM model was used for time-domain extrapolation. He [15] proposed a pot extrapolation method based on improved time-domain load expansion for the relatively stable cutting force load of NC machine tools in the process of constant speed cutting. When

the load shows obvious periodicity and stability, the time-domain extrapolation method will have a good effect. Wang [16] extrapolated the torque load of a tractor power takeoff device based on the pot model to obtain the whole life load spectrum of the device.

When the load is nonstationary, the calculation amount of time-domain extrapolation is huge and the extrapolation time is long. In contrast, rain flow matrix extrapolation reflects strong engineering application value because it is suitable for random loads. The common methods of rain flow matrix extrapolation are the parametric method and nonparametric method. The parametric rain flow extrapolation method includes parametric rain flow extrapolation based on mixed distribution and parametric rain flow extrapolation based on joint distribution. Geng [17] proposed a road load spectrum construction method of automobile wheels based on the mixed distribution probability model. When Li [18] studied the machining center with model vdf100, he established the mean distribution of the cutting force based on mixed Weibull distribution and comprehensively determined the optimal basic function by using a variety of fitting test indexes. Chen [19] used the parameter extrapolation method to extrapolate the cutting load of the machining center and compiled the two-dimensional cutting load spectrum. According to the working condition characteristics of the variable speed cutting of an NC lathe, he [20] uses Weibull distribution and mixed Weibull distribution, respectively, to fit the amplitude and mean value of the load. Through correlation analysis, the mean value and amplitude of the load are not independent of each other. The joint probability distribution function of the mean value and amplitude of load is established by a copula function, and the cutting force spectrum of the NC lathe is compiled. Zhai [21] aimed at the transmission system of a peanut combine harvester, established the joint distribution probability density function for load extrapolation and obtained the one-dimensional torque eight-stage program spectrum through dimension reduction. The parametric rain flow extrapolation method is suitable for situations where the signal is relatively stable and the shape of the rain flow matrix is relatively simple. However, the shape of the rain flow matrix of a load with strong randomness is complex. This method has the limitations of failing to fit or struggling to determine the distribution function.

Nonparametric extrapolation is based on the idea of kernel density estimation. It does not need to make assumptions about the estimated data in advance. It only needs to select the input data variables, the core function and the corresponding bandwidth to estimate the probability density function of the input data. The method of kernel density estimation is to study the whole distribution based on sample data [22]. In kernel density estimation, the choice of bandwidth has a great influence on the estimation accuracy [23]. Common methods for determining the optimal bandwidth include insertion based on the thumb rule, etc. Liu [24] firstly deduced the maximum of the amplitude based on the parameter distribution, and then established the kernel density estimate, calculated the bandwidth based on the insertion method, completed the mean and amplitude extrapolation at the same time and established the pavement load spectrum of the drive shaft of a military track car. Yu [25] carried out a load acquisition test based on user road big data research, calculated the bandwidth by the insertion method, established a two-dimensional kernel density estimation model, extrapolated the rain flow matrix of each road condition by Monte Carlo simulation and finally established the full-life load spectrum of a front shock absorber. In order to use different bandwidths for each data point, scholars put forward the idea of kernel density estimation of adaptive bandwidth. Yang [26] proposed a method for adjusting the local bandwidth based on Euclidean distance and maximum distance for the variable load characteristics of wind power systems with multiple wind farms. Niu [27] partitioned the data area based on the quadtree partition algorithm according to the load characteristics of the frame of a corn harvester, calculated the bandwidth separately by the thumb method for data of different areas and realized the kernel density estimation with different bandwidths in different data areas. Wan [28] obtained rain flow statistics on the loads of a loader working device, calculated the bandwidth with the thumb method and introduced an adaptive factor. The size of the adaptive factor was related to the density of

data points. A two-dimensional mathematical model for estimating kernel density with adaptive bandwidth was established. Finally, it was compared with the joint mathematical model for estimating the probability density of the parameter distribution. The results showed that the damage of the load spectrum established by the parameter distribution model is significantly smaller, while the fatigue life calculated by damage generated by the two-dimensional kernel density estimation model with adaptive bandwidth is closer to the design life of the device. Nonparametric rain flow extrapolation is applicable to loads with strong randomness and a complex shape of the rain flow matrix. In the field of automotive transmission, especially the special transmission assembly of new energy vehicles, the nonparametric extrapolation method in rain flow matrix extrapolation is more suitable because of its various and complex operation modes and variable driving conditions.

For the loads of different operation modes of PHEVs, the form of the rain flow matrix is quite complex, and the mean and amplitude of loads do not always conform to any distribution. At this time, the parameter rain flow method cannot fit the mean and amplitude of these loads very well. Based on the idea of kernel density estimation, instead of the hypothetical fitting of the distribution of the mean and amplitude of the load, the method of nonparametric rain flow extrapolation can estimate the probability density function of input data only by taking the mean and amplitude of the load as input variables and selecting the corresponding core function and bandwidth, which is suitable for loads with strong randomness and a complex shape of the rain flow matrix. In the field of automotive transmission, especially special transmission assemblies for new energy vehicles, the operation modes are various and complex, and the driving conditions are also variable. The distribution of load averages and amplitudes for some modes of operation does not conform to any existing distribution and cannot be assumed in advance. Nonparametric extrapolation can solve this problem very well. Therefore, the nonparametric rain flow extrapolation method of rain flow extrapolation is used to extrapolate the load spectrum of PHEV transmission.

The purpose of this paper is to compensate for the gap in compiling the load spectrum of a PHEV transmission assembly. Firstly, based on the research results on the driving habits of users in the national standard "Driving Test Method for Automotive Reliability" (GB/T 12678-2021), load acquisition test conditions are divided into urban road conditions, high-speed road conditions, provincial road conditions and poor road conditions. The load data acquisition is carried out in the ratio of 55:30:10:5 for the driving mileage of each road condition. The single acquisition cycle is 300 km and the total acquisition cycle is three times. As far as possible, the acquired test load should be correlated with the user operating conditions. By analyzing the hybrid structure and control strategy of the test vehicle, the load of the test vehicle can be classified and counted according to the operation mode, which can be divided into a test load in pure electric mode, test load in series mode, test load in engine direct drive mode and test load in parallel mode. The loads generated by the engine and the drive motor in each mode are separated and the four-point rain flow counting statistics are carried out to obtain the power source rain flow matrix and statistical histogram of each mode. The two-dimensional kernel density estimation method is used to estimate the kernel density. The optimum bandwidth is calculated based on the thumb rule to estimate the kernel density. An adaptive factor is introduced to adjust the bandwidth of the kernel density estimation. The second kernel density estimation is carried out. Then, the frequency extrapolation is completed by Monte Carlo simulation. The dimension of the extrapolated two-dimensional load spectrum is reduced by means of the variable mean method and Goodman mean stress correction method, and the one-dimensional load spectrum is obtained. Based on the principle of equal damage, the amplitude is divided into eight grades and the eight-stage program load spectrum of the transmission assembly is established. Finally, three-dimensional modeling of the transmission assembly and life simulation prediction based on the load spectrum are carried out in the Romax Designer environment. The number of cycles of load spectrum conditions required for transmission assembly failure is calculated by simulation, and

the fatigue life of the transmission assembly is deduced. With the statistical after-sales service data of this model from listing to the present as auxiliary proof, the failure mileage range of each gear of the transmission assembly under 99.5% confidence is calculated. The results of the simulation test are all within the confidence interval, which shows that the failure components and their failure mileage life in the simulation test results are basically consistent with the results of actual users, as confirmed using feedback.

The main contributions of this paper are as follows.

In this paper, the load data measurement test is designed by analyzing the hybrid power system of the test vehicle and the user data provided by the national standard. Different operating modes are fitted and extrapolated, respectively, and then the extrapolated rain flow matrix is converted into a one-dimensional load spectrum, and simulation fatigue tests are carried out based on the load spectrum effectiveness.

The structure of this paper is as follows: Section 2 introduces the basic parameters of the test vehicle and the hybrid power system, the method of data preprocessing, the two-dimensional kernel density estimation model, the nonparametric extrapolation method of the load spectrum and the compilation of the eight-level program load spectrum. In Section 3, the transmission assembly is modeled, and fatigue simulation tests are performed based on the compiled load spectrum. In Section 4, the results of the simulation are discussed according to the characteristics of the operating mode and are validated based on after-sales maintenance statistics. Finally, we present the conclusions.

## 2. Materials and Methods

### 2.1. Hybrid System and Control Strategy of Test Vehicle

The research object of this paper is a PHEV of a Chinese brand, and its power system parameters are shown in Table 1.

**Table 1.** The basic parameters of the test vehicle.

| Project | Parameter |
|---|---|
| Curb weight | 1660 kg |
| Length × width × height | 4765 mm × 1837 mm × 1495 mm |
| Engine maximum power | 81 kW |
| Engine maximum torque | 250 Nm/4500 rpm |
| Maximum power of drive motor | 240 Nm |
| Maximum torque of drive motor | 132 kW |
| Combined maximum torque | 490 Nm |
| Transmission | E-CVT |
| Front/rear wheel specifications | 225/60 R16 |
| Power battery capacity | 8.32 kWh |

The test vehicle is a torque-coupled hybrid electric vehicle. Its powertrain is mainly composed of a hybrid engine, drive motor, generator and transmission assembly. Its hybrid power system is shown in Figure 1.

The test vehicle is mainly driven by the engine and drive motor. The vehicle control unit (VCU) controls the engine management system (EMS) and motor control unit (MCU) to control the work of the engine and drive motor. In addition, the vehicle controller VCU also controls the battery management system (BMS), thereby controlling the charge and discharge of the battery. The data exchange of each controller is carried out on the controller area network (CAN) bus. The combination of the clutch can couple the power output from different power sources, couple the power of the engine with the power of the drive motor, output it to the reducer gear set through the intermediate shaft and finally drive the car to the drive axle. The combination and separation of the clutch will directly determine the transmission route of the mechanical flow of the transmission assembly.

When driving, the vehicle will switch among different operation modes according to the instructions given by the control strategy. Among them, there are four operation modes with the highest proportion of time, namely pure electric mode, series mode, engine

direct drive mode and parallel mode. The transmission assembly has different mechanical energy transmission routes under different operation modes. When the vehicle is in the low-speed state or starts, it switches to pure electric mode and makes full use of the external characteristics of the driving motor with high torque at low speed to drive the vehicle efficiently. When the vehicle is running at low speed and the state of charge (SOC) is too low, it switches to series mode. At this time, the engine will start to drive the generator to generate electricity and maintain the battery voltage and SOC, while the driving motor is still responsible for driving the whole vehicle. When the vehicle runs at a high speed, it is switched to parallel mode. At this time, the engine works in the optimal economic area, and the intermediate shaft of the transmission assembly couples the torque of the drive motor and the engine and outputs it to the reducer and drive axle end. When the vehicle runs at high speed, it is switched to direct engine drive mode, and the drive motor does not participate in the drive. At this time, the whole vehicle is directly driven by the engine. Through the preliminary analysis, it can be seen that under different operation modes, the number and size of power source load input borne by the transmission assembly are different.

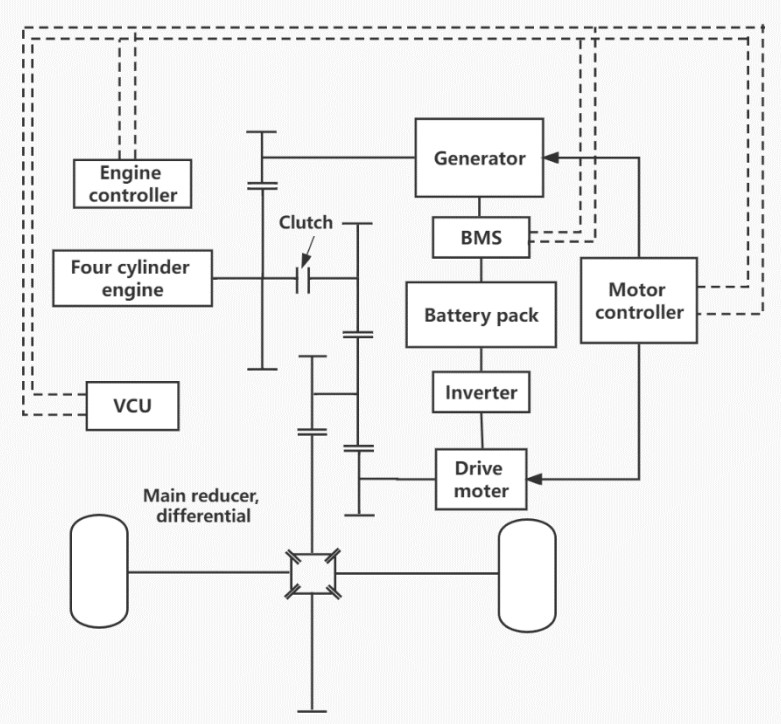

**Figure 1.** Test vehicle's hybrid system.

### 2.2. Data Acquisition of Transmission Assembly

Transmission load data are not only an important input of simulation in the transmission design stage, but also the input of the reliability bench test of the transmission assembly prototype in the later stage. The compilation data of the transmission load spectrum for the simulation analysis and bench test generally come from the road load data related to typical users and the fatigue durability test data of the automobile test field. The types, mileage and strengthening grade of various roads in the automobile test site are formulated by engineers through measuring some representative roads, and their values are fixed; they often deviate from the actual road conditions of users. The road load data related to typical users come from the real load borne by the transmission assembly during the actual use of users. For different road conditions, the load data have their own characteristics. In order for the compiled load spectrum to better reflect the damage caused by users in the actual use process, and to realize the damage caused by driving during the design life mileage,

which is consistent with the damage caused by the durability test in the automobile test field, this paper uses the road load data related to typical users.

Based on the user's big data investigation results in the Chinese national standard "automobile reliability driving test method" (GB/T 12678-2021), it is determined that the typical test conditions of the load acquisition test are urban road conditions, high-speed road conditions, provincial road conditions and poor road condition. The proportion of each road condition is 55:30:10:5, and the load data with a total mileage of 300 km are collected. In order to ensure the reliability of data and reduce accidental errors, three cycles are collected for each road condition in the test. With the increasingly mature calibration technology of the engine and motor, the deviation between engine torque, speed, motor torque and speed reflected by the CAN bus data signal and the real value is very small. Therefore, the CAN bus is selected for data acquisition. We read and record the signals of the ECU, VCU and BMS through the CAN bus, and convert the BLF format data into Excel format data in CANalyzer software. This paper mainly measures the key parameters in the transmission system, such as engine torque and speed, driving motor torque and speed, generator torque and speed, vehicle speed, gear signal and so on.

### 2.3. Data Preprocessing and Time-Domain Verification Analysis

Data preprocessing includes invalid load removal and deburring. A small-amplitude load has little contribution to the damage of parts, but its frequency is too high, which increases the amount of calculation in subsequent data processing. Therefore, this paper sets the threshold to 5% of the torque load range, and eliminates the small-amplitude load whose amplitude is less than this threshold.

In deburring, based on the PauTa criterion, if the absolute value of the difference between the load data point and the load mean is greater than 5 times the standard deviation, the point load is regarded as a burr point. The load data of adjacent normal points are given to the burr points to eliminate the burr.

As can be seen from Figure 2, the burrs of the load data points are well corrected after preprocessing. After the invalid amplitude is removed, the load time mileage is shortened, the data points are reduced, and the calculation amount of the subsequent load spectrum compilation is reduced.

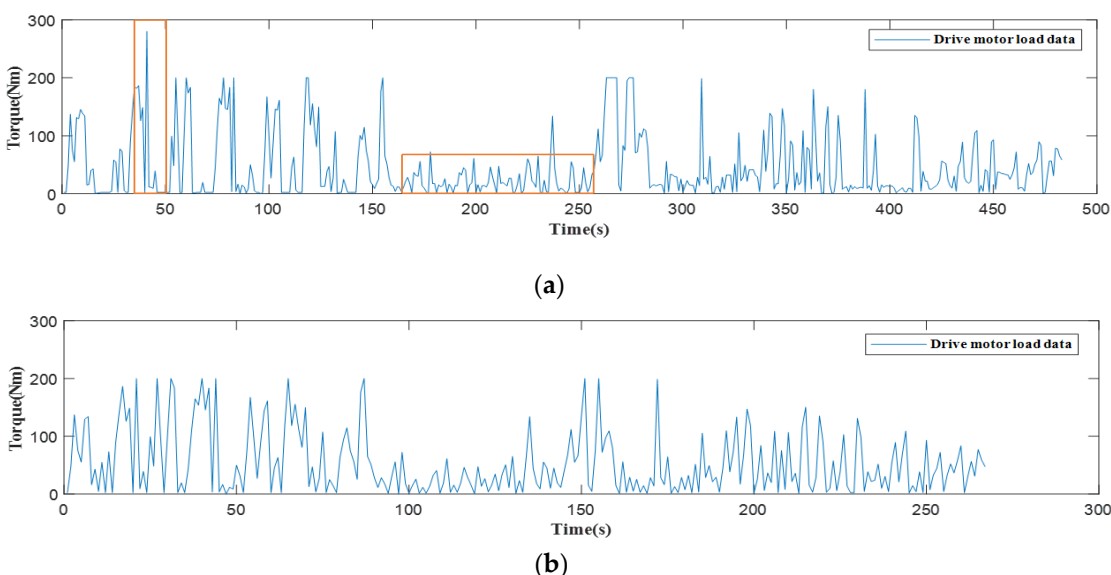

**Figure 2.** Comparison of partial load data before and after processing: (**a**) before preprocessing, (**b**) after preprocessing. In the orange box are the burr signal and the invalid amplitude signal.

The time-domain verification and analysis of the test signal is mainly used to analyze the waveform and statistical characteristics of the signal. The typical statistical characteris-

tics include the maximum value, mean value, standard deviation, etc., obtained through the analysis of the characteristics of each set of cyclic load data, and the repeatability of each set of cyclic load data can be tested. In addition, the coefficient of variation is introduced as the evaluation index of cyclostationarity. The statistical characteristic parameters and coefficient of variation of each cycle are shown in Table 2.

**Table 2.** Statistical characteristic parameters and coefficient of variation of each cycle.

| Statistical Parameters Under Different Working Modes | | Cycle 1 | Cycle 2 | Cycle 3 |
|---|---|---|---|---|
| Statistical parameters of driving motor torque in pure electric working mode | Maximum (Nm) | 201.30 | 204.58 | 202.31 |
| | Mean value (Nm) | 48.80 | 49.72 | 48.40 |
| | Standard deviation (Nm) | 51.12 | 50.82 | 50.41 |
| | Coefficient of variation | 0.95 | 0.98 | 0.96 |
| Torque statistical parameters of drive motor in series operation mode | Maximum (Nm) | 149.4 | 142.85 | 151.68 |
| | Mean value (Nm) | 51.31 | 50.29 | 52.87 |
| | Standard deviation | 32.55 | 31.28 | 33.49 |
| | Coefficient of variation | 1.57 | 1.60 | 1.58 |
| Statistical parameters of engine torque in series operation mode | Maximum (Nm) | 43.07 | 41.01 | 39.20 |
| | Mean value (Nm) | 14.91 | 13.78 | 13.63 |
| | Standard deviation (Nm) | 9.50 | 9.72 | 9.81 |
| | Coefficient of variation | 1.55 | 1.41 | 1.38 |
| Statistical parameters of engine torque in direct drive mode | Maximum (Nm) | 220.03 | 224.85 | 221.12 |
| | Mean value (Nm) | 67.04 | 66.71 | 65.51 |
| | Standard deviation (Nm) | 48.11 | 47.03 | 47.05 |
| | Coefficient of variation | 1.42 | 1.41 | 1.39 |
| Statistical parameters of driving motor torque in parallel operation mode | Maximum (Nm) | 201.11 | 204.17 | 199.03 |
| | Mean value (Nm) | 30.13 | 33.58 | 27.41 |
| | Standard deviation (Nm) | 36.21 | 35.41 | 34.33 |
| | Coefficient of variation | 0.82 | 0.94 | 0.79 |
| Statistical parameters of engine torque in parallel operation mode | Maximum (Nm) | 229.14 | 218.52 | 234.37 |
| | Mean value (Nm) | 79.52 | 76.50 | 81.41 |
| | Standard deviation (Nm) | 52.62 | 53.81 | 51.52 |
| | Coefficient of variation | 1.51 | 1.42 | 1.58 |

It can be seen from the above table that the statistical characteristic parameters of different cycles have little difference, and the difference between variation coefficients is also small, indicating that the discreteness of each load cycle under the same road condition is small and the repeatability is good.

*2.4. Determination of Extrapolated Load Samples*

In order to reduce the influence of random data, three cycles will be collected during the measured road load spectrum test and load data collection. Before the statistical counting of load data frequency, it is necessary to select a representative cyclic load as the data sample. The pseudo damage of the measured load spectrum can truly reflect the strength of the load spectrum. The measured load data with the largest total pseudo damage can be selected as the basic load data of the subsequent extrapolated load spectrum [29]. Therefore, the damage caused by different cyclic loads to parts can be used as the basis for the selection of the basic load data. However, in engineering applications, it is difficult to obtain the accurate S-N curve of the studied parts, and there is no need to calculate the accurate damage value caused by the load in the process of determining the load spectrum sample. Pseudo damage is usually used to evaluate the severity of the load [30,31]. In this paper, the pseudo damage caused by different cyclic loads will be used as the basis for the selection of sample data. The rain flow counting method is used to extract the information of each cycle in the measured load. Based on the Miner criterion and the fact that there is no strict corresponding relationship between pseudo damage and structural parameters, the calculation of pseudo damage $d_p$ is simplified as [32]

$$d_p = \sum_{i=1}^{z} S_i^k \tag{1}$$

where, $S_i$ is the amplitude of the load cycle, $K$ is the structural material parameter, and $z$ is the number of load cycles in the rain flow count. According to the pseudo damage calculation criterion of auto parts [14], the $k$ here is 5. The amount of pseudo damage produced by each cycle is shown in Table 3.

**Table 3.** Pseudo damage caused by each power source load in each cycle.

| Pseudo Damage Caused by Power Source Torque in Different Working Modes | Cycle 1 | Cycle 2 | Cycle 3 |
|---|---|---|---|
| Pseudo damage caused by torque of driving motor in pure electric working mode | $1.7002 \times 10^{13}$ | $1.6802 \times 10^{13}$ | $1.6582 \times 10^{13}$ |
| Pseudo damage caused by engine torque in series mode | $1.2877 \times 10^{12}$ | $1.1943 \times 10^{12}$ | $1.2754 \times 10^{12}$ |
| Pseudo damage caused by torque of driving motor in series mode | $2.7292 \times 10^{9}$ | $2.6898 \times 10^{9}$ | $2.7188 \times 10^{9}$ |
| Pseudo damage caused by engine torque in direct drive mode | $1.0863 \times 10^{13}$ | $1.0554 \times 10^{13}$ | $1.0631 \times 10^{13}$ |
| Pseudo damage caused by engine torque in parallel mode | $1.2161 \times 10^{13}$ | $1.1321 \times 10^{13}$ | $1.1985 \times 10^{13}$ |
| Pseudo damage caused by torque of driving motor in parallel mode | $4.1232 \times 10^{13}$ | $4.0211 \times 10^{13}$ | $4.1119 \times 10^{13}$ |

It can be seen from Table 3 that the pseudo damage amount corresponding to cycle 1 is the largest. According to the maximum damage principle, this cycle is selected as the sample load data for subsequent data processing.

*2.5. Statistical Counting of Load Sample Data*

Based on the characteristic signal of the PHEV operation mode, the load data are separated and extracted according to different operation modes and power sources, and the torque and speed of the driving motor in pure electric mode, speed and torque of the engine and drive motor in series mode, engine speed and torque in direct drive mode and speed and torque of the engine and drive motor in parallel mode are extracted. The four-point rain flow counting method is adopted to count the above load data and count the mean value and amplitude of the load data. In order to reduce the error caused by load amplitude and mean value classification, the amplitude load classification should not be lower than 30 levels [33]. Therefore, this paper uses a $32 \times 32$ matrix to save the average amplitude frequency information of the load data, and the statistical histogram is shown in Figure 3.

*2.6. Adaptive Bandwidth Two-Dimensional Kernel Density Estimation Model Establishment*

In order to ensure that the load spectrum is more in line with the actual situation, it is necessary to extrapolate the load frequency before compiling the load spectrum. According to the previous discussion, when the data distribution of the rain flow counting matrix is random and it is difficult to describe its distribution by parameter estimation, the nonparametric estimation method is generally used. In this paper, the nonparametric extrapolation method based on kernel density estimation is used to extrapolate the load frequency. This method can estimate the overall distribution law according to the sample, and then expand the extrapolation of limited samples according to the overall distribution law. The kernel density estimation method can retain the distribution law of the data themselves without making any assumptions about the data. It only needs to determine the output data variables and select the kernel function and bandwidth to estimate the probability density function of the output data [34]. The data extrapolation of the rain flow counting matrix is a two-dimensional problem, so the two-dimensional kernel density extrapolation method is adopted. The two-dimensional kernel density estimation formula is

$$\hat{f}(x,y) = \frac{1}{nh_xh_y} \sum_{i=1}^{n} K\left( \left| \frac{x - x_i}{h_x} \right|, \left| \frac{y - y_i}{h_y} \right| \right) \tag{2}$$

$K(\cdot)$ needs to meet the following conditions:

$$\begin{cases} K(\cdot) \geq 0 \\ \iint K(\cdot)dx = 1 \end{cases}$$

where *n* is the number of input data points; $h_x$, $h_y$ are the bandwidth of the estimated load cycle amplitude and mean value for the kernel density; $x_i$, $y_i$ are the input amplitude and mean value of the load cycle; $K(\cdot)$ is the kernel function.

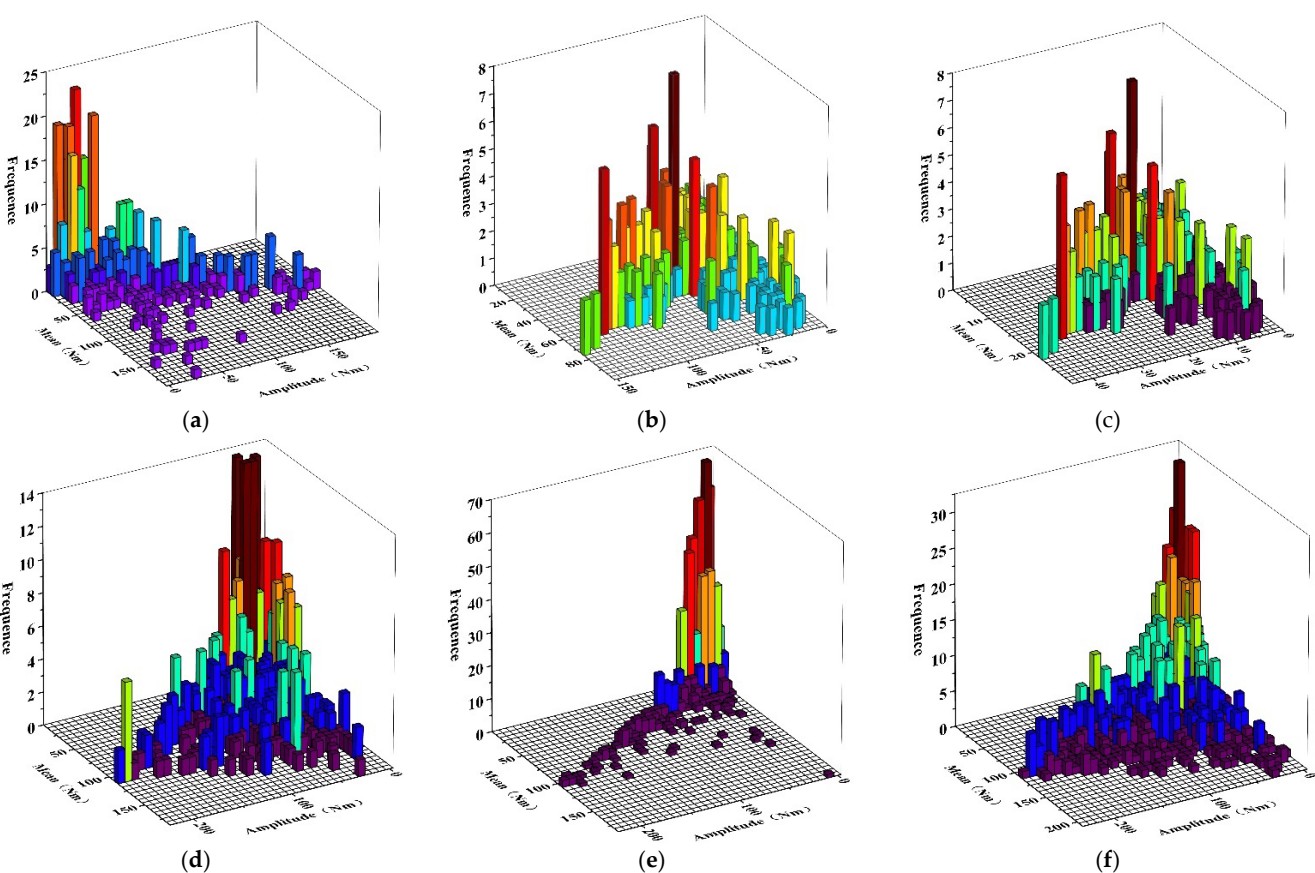

**Figure 3.** Load statistical histogram of each operation mode: (**a**) driving motor load in pure electric mode, (**b**) drive motor load in series mode, (**c**) engine load in series mode, (**d**) engine load in engine drive mode, (**e**) drive motor load in parallel mode, (**f**) engine load in parallel mode.

Choosing the appropriate kernel function and bandwidth, we can approach the real probability density function of any random variable without limitation. The most common types of kernel functions include the Gaussian kernel function, triangular kernel function, box kernel function and Epanechnikov kernel function. The form of kernel function has little impact on the accuracy of probability density estimation. Therefore, the smooth and continuous Gaussian kernel function is adopted in this paper. The expression of the two-dimensional Gaussian function is

$$K\left(\left|\frac{x-x_i}{h_x}\right|, \left|\frac{y-y_i}{h_y}\right|\right) = \frac{1}{2\pi} \exp\left(-\frac{(x-x_i)^2}{2h_x^2} - \frac{(y-y_i)^2}{2h_y^2}\right) \tag{3}$$

Compared with the type of kernel function, the choice of bandwidth has a greater impact on the accuracy of kernel density estimation. The bandwidth determines the smoothness of kernel density estimation. The larger the bandwidth, the smoother the kernel density estimation curve, but there will be a large deviation from the actual probability density curve; if the bandwidth is too small, the kernel density estimation curve will be

very steep, and there will also be a large deviation. At present, the most widely used calculation method for the optimal bandwidth is the thumb method, and its calculation formula is

$$h = \left( \frac{4}{d+2} \right)^{\frac{1}{d+4}} \sigma n^{-\frac{1}{d+4}} \tag{4}$$

where $d$ is the dimension of kernel density estimation, taken as 2; $\sigma$ is the standard deviation of the input two-dimensional data sample; $n$ is the number of data points.

However, each data point adopts the same bandwidth for kernel density estimation, and the results are not satisfactory. Therefore, some scholars proposed [35,36] a kernel density estimation method by introducing a bandwidth adaptive factor, so that each data point adopts a different bandwidth. The mathematical expression of the two-dimensional kernel density estimation of adaptive bandwidth is as follows:

$$f(x,y) = \frac{1}{n} \sum_{i=1}^{n} \left[ \frac{1}{h_x h_y \lambda_i^2} K \left( \frac{x - x_i}{h_x \lambda_i}, \frac{y - y_i}{h_y \lambda_i} \right) \right] \tag{5}$$

where $\lambda_i$ is the adaptive factor of data points, and its calculation method is as follows:

$$\lambda_i = \left[ \frac{p(x_i, y_i)}{g} \right]^a \tag{6}$$

where $a$ is the sensitivity parameter, taken as $-0.5$.

$$p(x_i, y_i) = \frac{c_i}{\sum\limits_{i=1}^{n} c_i} \tag{7}$$

$$\lg(g) = \frac{1}{n} \sum_{i=1}^{n} \lg[p(x_i, y_i)] \tag{8}$$

where $c_i$ is the number of occurrences of $(x_i, y_i)$ in the rain flow counting matrix.

In this paper, the optimal bandwidth is calculated based on the thumb rule for preliminary kernel density estimation, and then the adaptive factor is solved for the second kernel density estimation. Finally, the probability density function is obtained for Monte Carlo simulation, and the extrapolated rain flow matrix is obtained, as shown in Figure 4.

Using the nonparametric kernel density estimation model, we not only hope that the established model can fit the distribution of data, but also hope that, through the fitting of the distribution, we can predict some random loads that are not measured in the actual test. It can be seen that the rain flow matrix extrapolation using the nonparametric kernel density estimation model can not only extrapolate the measured load, but also extrapolate some loads that do not appear in the test. The mean value and amplitude of the load are extrapolated synchronously, which well reflects the variability of the load under complex working conditions. The fatigue life prediction based on the results is closer to the real situation.

### 2.7. Establishment of Eight-Stage Program Load Spectrum

The extrapolated load spectrum is the load spectrum in the form of the rain flow matrix, which cannot be directly used as the input of the bench test or simulation test. In this paper, the variable mean method is used to convert the two-dimensional load spectrum into the load spectrum with unified mean. The conversion formula is as follows:

$$M_i = \frac{\sum\limits_{j}^{32} M_j n_{ij}}{\sum\limits_{}^{32} n_{ij}} \tag{9}$$

where $M_i$ is the equivalent mean value of the load mean value corresponding to the *i*-th amplitude; $M_j$ is the average value of the *j*-th load; $n_{ij}$ is the frequency of level *i*-th load amplitude and level *j*-th load mean.

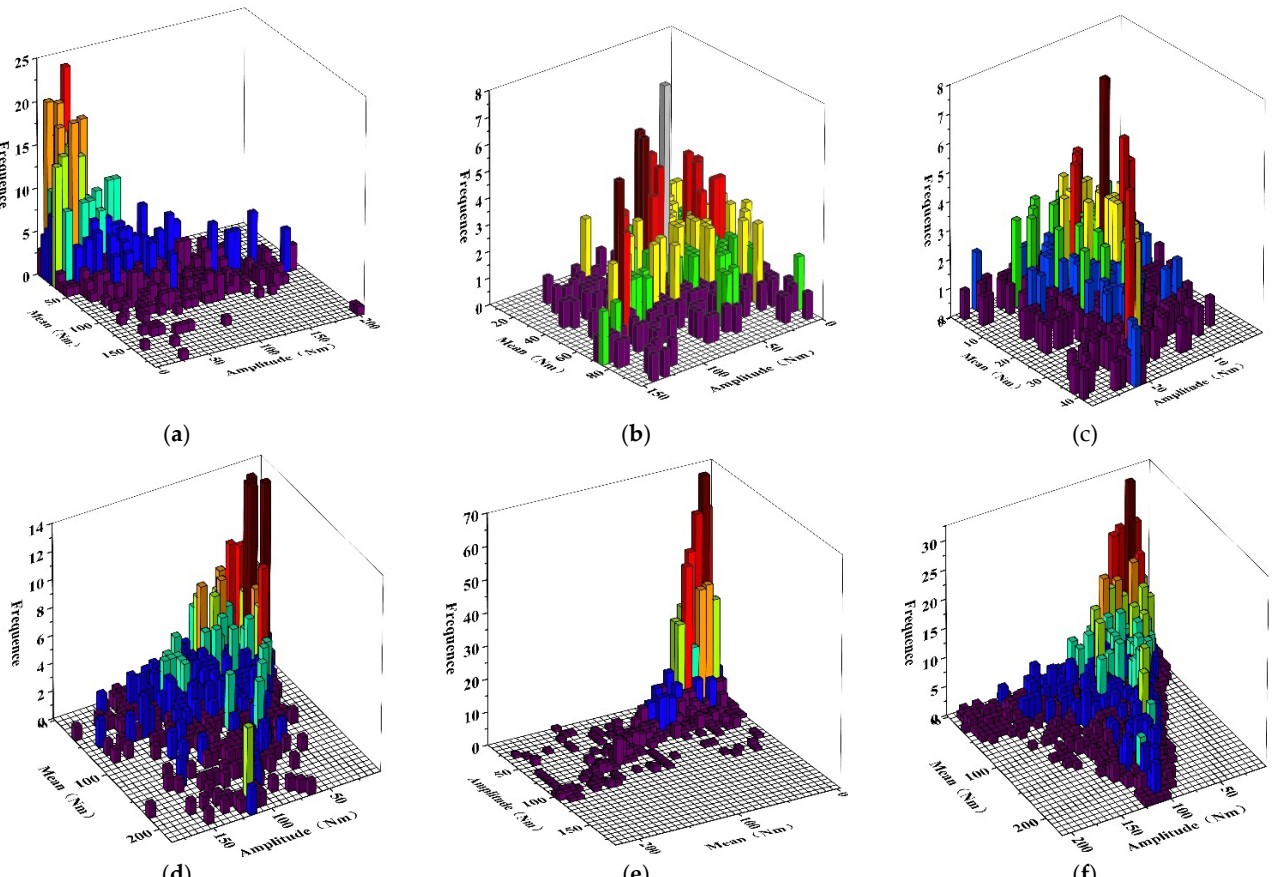

**Figure 4.** Histogram of load extrapolation of each mode: (**a**) driving motor load in pure electric mode, (**b**) drive motor load in series mode, (**c**) engine load in series mode, (**d**) engine load in engine drive mode, (**e**) drive motor load in parallel mode, (**f**) engine load in parallel mode.

Then, the average stress is corrected by the Goodman equation, and the two-dimensional load spectrum is transformed into a one-dimensional load spectrum with only amplitude as the variable. The conversion formula is as follows:

$$T_e = \frac{T_a}{1 - \frac{T_m}{T_b}} \tag{10}$$

where $T_e$ is the torque amplitude of the symmetrical cyclic load, which is equivalent to the load state under the simultaneous action of torque amplitude $T_a$ and torque mean value $T_m$, and $T_b$ is the torque strength limit, which is calculated from the material and size of the transmission shaft.

Finally, based on the equal damage principle, the one-dimensional load spectrum is transformed into the common eight-stage program load spectrum. The calculation formula is as follows:

$$N_{eq} = \sum \left( S_j / S_{eq} \right)^m \cdot N_j \tag{11}$$

where $N_{eq}$ is the equivalent cycle frequency; $S_{eq}$ is the load value to be converted; *m* is the power exponent of the S-N curve, which is determined by the material properties; $N_j$ is the number of cycles under corresponding loads at all levels; $S_j$ is the *j*-th amplitude after correction; *j* represents the load spectrum series. Table 4 shows the eight-stage program load spectrum of each operation mode.

**Table 4.** Eight-stage program load spectrum.

| Operation Mode | Loading Power Source | Range (Nm) | Frequency | Operation Mode | Loading Power Source | Range (Nm) | Frequency |
|---|---|---|---|---|---|---|---|
| Pure electric mode | Drive motor | 200 | 1 | Engine direct drive mode | Engine | 216 | 2 |
| | | 168 | 5 | | | 188 | 7 |
| | | 135 | 38 | | | 156 | 41 |
| | | 110 | 90 | | | 122 | 259 |
| | | 86 | 380 | | | 95 | 420 |
| | | 59 | 2676 | | | 64 | 3631 |
| | | 34 | 32,885 | | | 38 | 29,976 |
| | | 15 | 70,232 | | | 16 | 60,218 |
| Series mode | Drive motor | 147 | 4 | Parallel mode | Drive motor | 200 | 1 |
| | | 126 | 16 | | | 171 | 12 |
| | | 107 | 25 | | | 151 | 53 |
| | | 88 | 75 | | | 104 | 228 |
| | | 68 | 160 | | | 80 | 538 |
| | | 47 | 694 | | | 58 | 2015 |
| | | 26 | 18,221 | | | 34 | 27,127 |
| | | 10 | 21,384 | | | 13 | 48,238 |
| | Engine | 42 | 6 | | Engine | 226 | 4 |
| | | 36 | 27 | | | 190 | 21 |
| | | 31 | 31 | | | 157 | 141 |
| | | 24 | 404 | | | 126 | 789 |
| | | 19 | 479 | | | 97 | 2611 |
| | | 13 | 2667 | | | 68 | 10,481 |
| | | 8 | 4883 | | | 39 | 20,281 |
| | | 3 | 10,792 | | | 13 | 50,625 |

According to the eight-stage program loading spectrum shown in the Table 4, we can determine the corresponding frequency of loads at all stages and calculate the loading time of loads at all stages according to the running time of each mode. The speed of loading is determined by the average speed change of the power source in this mode. The settings are shown in Table 5.

**Table 5.** Load speed setting.

| Operation Mode | Power Source | Loading Speed (rpm) |
|---|---|---|
| Pure electric mode | Drive motor | 8500 |
| Engine direct drive mode | Engine | 3000 |
| Series mode | Drive motor | 9600 |
| | Engine | 1200 |
| Parallel mode | Drive motor | 12,000 |
| | Engine | 3200 |

## 3. Load Spectrum Simulation Application

### 3.1. Establishment of Simulation Model

In order to predict the life of parts in advance, based on the eight-stage program load spectrum that can be used for bench fatigue test input, this section models the transmission assembly mechanism used in the test under the environment of Romax Designer simulation software(R18, Nottingham, UK), and then takes the load spectrum of each operation mode as the power input of the simulation model to simulate and analyze the transmission assembly to obtain the damage and life of transmission parts.

Based on the real structure and dimensional parameters of the transmission assembly of the test vehicle, we input the length, inner diameter, outer diameter and material of each shaft; set the normal modulus, pressure angle, helix angle, number of teeth, tooth width, material, web width, tooth top height and tooth root height of each pair of gears; model

and assemble each shafting of the transmission assembly; and obtain the geometric model of the transmission assembly. Figure 5 shows the final transmission assembly. Pinion1 is the pinion on the generator shaft and wheel1 is the large gear engaged with it. The gear sets meshed with the engine shaft, intermediate shaft and drive motor shaft are Wheel2, Pinion2-1 and Pinion2-2, respectively; The gear sets meshed between the intermediate shaft and the main reducer are Wheel3 and Pinion3, respectively. The dark blue parts on each shaft are rigid bearings, and the orange-yellow components on the engine shaft are clutches. Wheel2 on the engine shaft is connected with the shaft through bearings. When the clutch is disconnected, Wheel2 is in an idle state, and when the clutch is engaged, Wheel2 is locked with the engine shaft.

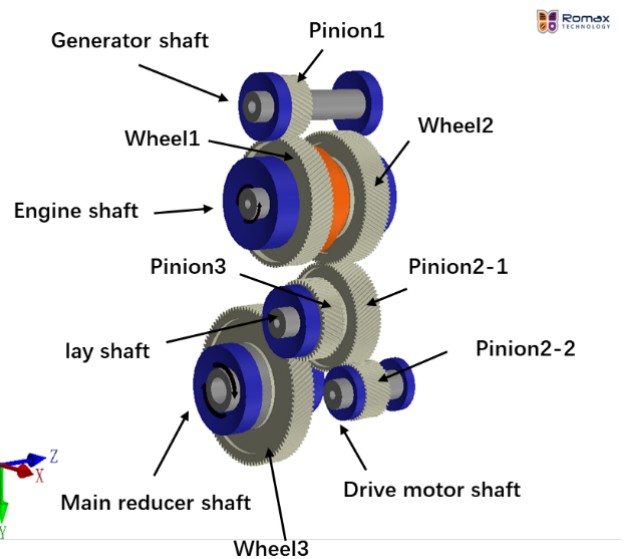

**Figure 5.** Simulation model of transmission assembly.

### 3.2. Simulation Settings

Regarding the load spectrum simulation loading setting, in pure electric mode, the drive motor shaft loads the drive motor torque load, the clutch on the engine shaft is disconnected, and the engine does not run, as shown in Figure 6a. In series mode, the drive motor shaft loads the drive motor torque load, the clutch on the engine shaft is disconnected, and the engine shaft loads the engine load torque. At this time, the engine only drives the generator gear to rotate, as shown in Figure 6b. In parallel mode, the clutch on the engine shaft is engaged, the drive motor shaft loads the drive motor torque load, the engine shaft loads the engine torque load, and the intermediate shaft couples the two torques and outputs them to the reducer, as shown in Figure 6c. In the direct drive mode of the engine, the drive motor does not participate in the drive. The engine shaft loads the engine torque load and outputs it to the reducer through the intermediate shaft, as shown in Figure 6d. Dark red indicates the gear loading surface, while light red indicates the gear loading surface. We set the lubricating oil as ISO vg100 and the temperature as 70 °C, and take the load spectrum of each operation mode as the input power flow of the model.

### 3.3. Simulation Results

The load spectrum of the four operation modes is simulated and loaded according to the above load spectrum, which is set in the Romax Designer simulation software environment to form a working cycle with a driving mileage of 300 km and loaded for operation. Through simulation analysis, the number of working cycles that can be loaded by each gear can be obtained, and then the service life mileage of the transmission assembly can be inferred. The simulation results are shown in Table 6.

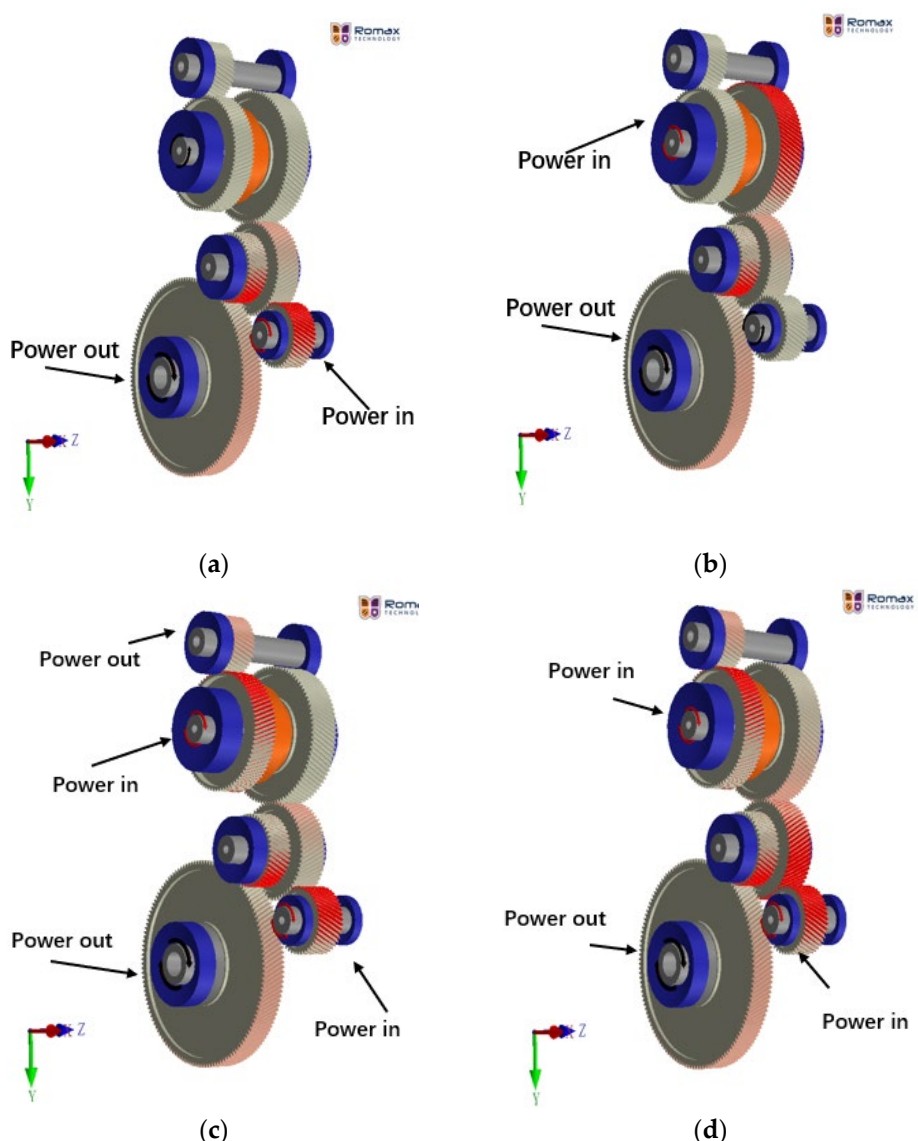

**Figure 6.** Schematic diagram of power flow input and output in each mode: (**a**) pure electric mode, (**b**) engine direct drive mode, (**c**) series mode, (**d**) parallel mode.

**Table 6.** Results of simulated mileage life of each gear.

| Gear | Number of Cycles Required to Break | Service Life Mileage /km |
|---|---|---|
| Wheel1 | 2295 | 688,500 |
| Pinion1 | 2299 | 689,700 |
| Wheel2 | 2332 | 699,600 |
| Pinion2-1 | 2271 | 681,300 |
| Pinion2-2 | 2152 | 645,600 |
| Wheel3 | 2133 | 639,900 |
| Pinion3 | 2208 | 662,400 |

## 4. Discussion

Through fatigue simulation analysis, in the transmission assembly, it is found that the gear pair of the main reducer first experiences fatigue failure, and then the gear pair on the intermediate shaft and the drive motor shaft experiences fatigue failure. The gears between the engine shaft and the generator shaft, as well as the large gears on the engine shaft, finally fail. This is due to the following reasons: firstly, regardless of the operation mode that the car is in, the meshing gear pair of the main reducer is in a working state; secondly,

the test vehicle is mainly driven by electricity, and the pure electric mode accounts for a large proportion. The working time of the meshing gear set on the intermediate shaft and the drive motor shaft is also relatively long. In addition to bearing the load of the drive motor in pure electric mode, the gear on the intermediate shaft also bears the load of the engine in parallel mode, which accelerates its fatigue failure. Finally, the driving time of the vehicle in series mode is relatively short, so the fatigue damage speed of the gear set on the engine shaft and generator shaft is the slowest. The design life of the transmission assembly of the test vehicle is 600,000 km, and all gear parts meet the design requirements.

Taking the after-sales maintenance data of the car from listing to the present as auxiliary proof, and taking Wheel1 as an example, a total of 189 failures have occurred in Wheel1 in the transmission assembly from listing to the present, and the average failure mileage is 627,958 km. Taking the failure data as a sample, when the confidence level is 99.5%, the confidence interval is calculated as 565,162.2–690,753.8 km. It can be considered that 99.5% of the Wheel1 failure mileage is distributed between 565,162.2 km and 690,753.8 km. Similarly, based on the maintenance statistics of other parts, the confidence interval of other parts with a confidence level of 99.5% can be calculated, as shown in Table 7.

**Table 7.** After-sales sample number and confidence interval of mileage.

| Gear | Simulation Results/km | Number of Samples | Confidence Interval /km |
|---|---|---|---|
| Wheel1 | 688,500 | 189 | 565,162.2–690,753.8 |
| Pinion1 | 689,700 | 114 | 574,847.1–702,590.9 |
| Wheel2 | 699,600 | 188 | 630,884.7–771,081.3 |
| Pinion2-1 | 681,300 | 152 | 573,489.9–700,932.1 |
| Pinion2-2 | 645,600 | 98 | 576,282.6–704,345.4 |
| Wheel3 | 639,900 | 128 | 550,152–672,408 |
| Pinion3 | 662,400 | 177 | 5,682,743.1–6,945,574.9 |

The life mileage of each part calculated from the simulation results is within the confidence interval, indicating that the faulty parts and their fault mileage life in the simulation test results are basically consistent with the results fed back by the actual users. To summarize, the method for compiling the load spectrum of the PHEV transmission assembly proposed in this paper has high accuracy and authenticity.

## 5. Conclusions

Aiming at addressing the research gap in the compilation of the load spectrum of the transmission assemblies of hybrid electric vehicle, research has been carried out.

Based on the control strategy of a hybrid electric vehicle, the route change of the total success rate flow transmission under different operation modes is analyzed, and it is clear that the compilation of the hybrid electric vehicle load spectrum needs to divide the load spectrum blocks of different operation modes based on different operation modes. In order to ensure that the compiled load spectrum reflects the damage caused by users in the actual use process, based on the big data investigation of users' driving conditions in Chinese national standards, the road load data acquisition test related to typical users is designed, and the typical test conditions collected are determined to be urban road conditions, high-speed road conditions, provincial road conditions and poor road conditions. The proportion of various working conditions is 55:30:10:5, the total acquisition mileage is 300 km, and the acquisition requires three cycles.

In order to reduce the influence of a high-frequency low-amplitude load on the calculation amount, we set the threshold to 5% of the load range, eliminate the small-amplitude load and deburr based on the principle of PauTa. By analyzing the statistical characteristics and variation coefficients of three cyclic loads, it is verified that the data have good repeatability, and the time-domain verification and analysis of the data are completed. Based on the principle of maximum damage, the cycle with the largest pseudo damage is selected as the sample data for load spectrum extrapolation. Based on the characteristic signals of each

operation mode, the load data are separated according to different operation modes and power sources. The rain flow statistics are carried out, respectively, and the 32 by 32 matrix is used to save the load information.

We select the Gaussian kernel function, calculate the bandwidth by using the thumb rule, preliminarily estimate the first kernel density and calculate the adaptive factor, establish the kernel density estimation mathematical model of the adaptive bandwidth and then estimate the kernel density of the average amplitude of the load. The estimated probability density function is simulated by the Monte Carlo method to obtain the extrapolated rain flow matrix. The average stress is modified by the variable mean method and Goodman equation, and the two-dimensional load spectrum in the form of a rain flow matrix is changed into a one-dimensional load spectrum with a single amplitude variable. Then, through the equal damage principle, it is transformed into an eight-stage program load spectrum.

Finally, in the Romax Designer simulation software environment, based on the real structure and size parameters of the transmission assembly of the test vehicle, the transmission assembly used in the test is modeled, and the simulation loading conditions are set according to the power flow characteristics of the transmission assembly under different operation modes. In the Romax Designer simulation software environment, it is combined into a load spectrum condition of 300 km driving mileage. According to the synthesized load spectrum conditions, the number of load spectrum condition cycles that can be loaded by each gear can be obtained, and then the service life mileage of the transmission assembly can be inferred. Taking the after-sales statistical maintenance data of this model from listing to the present as auxiliary proof, the failure mileage interval of each gear of the transmission assembly under the confidence of 99.5% is calculated. The simulation results are as follows: the mileage lives of Wheel1, Pinion1, Wheel2, Pinion2-1, Pinion2-2, Wheel3 and Pinion3 are 688,500 km, 689,700 km, 699,600 km, 681,300 km, 645,600 km, 639,900 km and 66,240 km, respectively. The results of the simulation test are all within the confidence interval, indicating that the failure components and the failure mileage life of the simulation test results are basically consistent with the results of the actual user feedback.

In this paper, a method of compiling the load spectrum is proposed, which designs the load data test associated with user habits and completes the extrapolation of the two-dimensional load spectrum and the compilation of the eight-level program spectrum. Based on the analysis of the simulation results, the durability of the transmission assembly can be predicted, forming a complete set of load spectrum compilation processes and reliability simulation test application methods. To summarize, the two-dimensional kernel density estimation model with adaptive bandwidth can well fit and extrapolate the hybrid modes of hybrid electric vehicles. The fatigue simulation analysis based on the load spectrum compiled by this extrapolation method is basically consistent with the feedback results of actual users. Therefore, the load spectrum of PHEV transmission compiled by using the extrapolation method of the two-dimensional kernel density estimation model with adaptive bandwidth has high accuracy. The compilation process of the load spectrum proposed in this paper has certain engineering guidance significance.

In the future, according to the characteristics of BEV single power source and single operation mode, the load spectrum extrapolation achieved by referring to the adaptive bandwidth kernel density estimation model used in this paper should also make it possible to fit the BEV load rain flow matrix better. Similarly, by modeling the BEV transmission assembly and setting the power flow in accordance with the operating mode, the fatigue life simulation analysis of the BEV based on the load spectrum should also be achieved. It would be of great significance for the future development of new energy automobiles for enterprises to form a system of load spectrum compilation processes and to establish an open database of compiled load spectrum data.

**Author Contributions:** Conceptualization, B.M. and W.W.; methodology, C.H.; software, C.H.; validation, Z.Z. (Zefeng Zheng) and Z.Z. (Zhiheng Zeng); formal analysis, Z.H.; investigation, B.M.; resources, W.W.; data curation, writing—original draft preparation, B.M.; writing—review and

editing, B.M.; project administration and supervision, W.W.; funding acquisition, C.W. All authors have read and agreed to the published version of the manuscript.

**Funding:** This research was funded by the R&D Program in Key Fields of Guangdong Province— R&D and Application of Key Technologies for New Energy Vehicle Gearboxes with High Efficiency, High Precision, Long Life and Low Noise (2020B090926004), Guangdong Province Key Field R&D Program—Electric Vehicle Powertrain Design and Optimization Industrial Software (2021B0101220003).

**Institutional Review Board Statement:** Not applicable.

**Informed Consent Statement:** Not applicable.

**Data Availability Statement:** Not applicable.

**Conflicts of Interest:** The authors declare no conflict of interest.

## Nomenclature

| | |
|---|---|
| BEV | battery electric vehicle |
| PHEV | plug-in hybrid electric vehicle |
| VCU | vehicle control unit |
| EMS | engine management system |
| MCU | motor control unit |
| CAN | controller area network |
| BMS | battery management system |
| SOC | state of charge |
| $d_p$ | pseudo damage |
| $S_i$ | the amplitude of the load cycle |
| $h_x, h_y$ | the bandwidth of the estimated load cycle amplitude and mean value for the kernel density |
| $x_i, y_i$ | the input amplitude and mean value of the load cycle |
| $K(\cdot)$ | the kernel function |
| $d$ | the dimension of kernel density estimation |
| $\sigma$ | the standard deviation of the input two-dimensional data sample |
| $n$ | the number of data points |
| $\lambda_i$ | the adaptive factor of data points |
| $a$ | the sensitivity parameter |
| $c_i$ | the number of occurrences of $(x_i, y_i)$ in the rain flow counting matrix |
| $M_i$ | the equivalent mean value of the load mean value corresponding to the $i$-th amplitude |
| $M_j$ | the average value of the $j$-th load |
| $n_{ij}$ | the frequency of level $i$-th load amplitude and level $j$-th load mean |
| $T_e$ | the torque amplitude of symmetrical cyclic load |
| $T_b$ | the torque strength limit |
| $N_{eq}$ | the equivalent cycle frequency |
| $S_{eq}$ | the load value to be converted |
| $m$ | the power exponent of the S-N curve |
| $S_j$ | the $j$-th amplitude after correction |

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
