# Peer review of "Compilation of Load Spectrum of PHEV Transmission Assembly and Its Simulation Application"

_machines, doi:10.3390/machines10070578_

Round 1
Reviewer 1 Report
-The legend in figure 3, must be in English
-authors talk about a variable à which take -0.5, please identify it in which equation
-the discussion section must be largely extended and try to show clearly the contribution
-correct figure 6, sub figures call.
- why this study is nominated for PHEV, I’m not sure, maybe it can be adapted for any electric vehicle. Authors must discuss this approach in future endeover new section
Author Response
Response to Reviewer 1 Comments
Point 1: The legend in figure 3, must be in English
Response 1: Thank you for your comment, we have reworked Figure 3.
Point 2: authors talk about a variable à which take -0.5, please identify it in which equation
Response 2: Thank you for your comment, we have identify it in equation 6.
Point 3: the discussion section must be largely extended and try to show clearly the contribution .
Response 3: Thank you for your comment, we have extened and rewritten paragraphs 1, 2, 3, 4, 5 and 7 of the discussion section.
Point 4: correct figure 6, sub figures call.
Response 4: Thank you for your comment, we have corrected Figure 6,sub figures call.
Point 5: why this study is nominated for PHEV, I’m not sure, maybe it can be adapted for any electric vehicle. Authors must discuss this approach in future endeover new section
Response 5: Thank you for your comment, bcause the research object of this paper and the fund project of this paper require our research team to study PHEV, the load spectrum compilation method in this paper is more inclined to PHEV. In addition, it can also adapt the method proposed in this paper according to the different research objects of scholars, and apply it to other electric vehicles. The following discussion is added to the last paragraph of the conclusion:
In the future, according to the characteristics of BEV single power source and single operation mode, the load spectrum extrapolation by referring to the adaptive bandwidth kernel density estimation model used in this paper should also make it possible to fit the BEV load rain flow matrix better. Similarly, by modeling the BEV transmission assembly and setting power flow in accordance with the operating mode, the fatigue life simulation analysis of the BEV based on the load spectrum should also be achieved. It is of great significance for the future development of new energy automobiles for enterprises to form a system of load spectrum compilation process and to establish an open database of load spectrum data compiled.
Reviewer 2 Report
In this paper, a method for compiling the load spectrum of the transmission assembly of plug-in hybrid electric vehicle is presented. The paper is written in good format but there are some basic ambiguities and it needs some major revisions. Some comments are as follows:
1- There is a similar paper (already published) to this manuscript:
"Load Spectrum Compilation Method of Hybrid Electric Vehicle Reducers Based on Multi-Criteria Decision Making"
What are the differents?
2- What is the main novelty of paper? it is better to clearly mentioned in the abstract and introduction.
3- There are some different fonts in the text. please correct them based on template.
4- What are the findings of the papers? mention them in short in abstract.
5- Eliminate lumped references and explain them.
6- It is better to have nomenclature part due to the large numbers of abbreviations.
7-The motivation part in the introduction is weak. After explaining different prametric and nonparametric rain flow methods, the weak points and disadvantages must be mentioned and then the advantages of proposed method must be highlighted.
8- There are some grammatical and typo errors. Please check the paper using a native spoken/software.
9-The last paragraph of introduction must be dedicated to paper structure explanation.
10- Mentioning tested or simulated parameters in Methodology section is not right, it is better to move them to the simulation section.
11- There is no explanation about Fig.1 inside the text!
12- Use the alignment for the rows of paragraphs.
13- In Fig. 2, it is better to put a b for each paper and mention which one is after or before case.
14- The fonts and the phase of written words in Fig.3 axes are hard to be realized. Please make them more clear.
15- Define all parameters in the equations, (for example "Landa" in eq.6)
16- The font size of Eq. 10 and 11 muse be decreased.
17- The discussion and explanation about obtained results and figures are very short. They need to be improved.
18- Also please clearly mention the main findings of paper even in quantitative mode in conclusion section.
Author Response
Response to Reviewer 2 Comments
[Comments and Suggestions for Authors]: In this paper, a method for compiling the load spectrum of the transmission assembly of plug-in hybrid electric vehicle is presented. The paper is written in good format but there are some basic ambiguities and it needs some major revisions.
Response: We sincerely appreciate the Reviewer #2’s approval of our work, having much review work and constructive comments on our manuscript. It was your valuable and insightful comments that led to possible improvements in the current version. Based on your valuable advice, this article have been carefully revised. The responses are stated as follows in red.
Point 1: There is a similar paper (already published) to this manuscript:
"Load Spectrum Compilation Method of Hybrid Electric Vehicle Reducers Based on Multi-Criteria Decision Making"
What are the differents?
Response 1: Thank you for your comment, paper, Load Spectrum Compilation Method of Hybrid Electric Vehicle Reducers Based on Multi-Criteria Decision Making, is studied on the main reducer of a 6-speed DCT hybrid electric vehicle, and the load spectrum of the main reducer is compiled. The decision-making algorithm is used to calculate the extrapolated samples required for load and the kernel density estimation model is established by using the Gauss core function to extrapolate the load spectrum of a single component of the final drive. However, for the key content in the kernel density estimation model, there is a lack of corresponding explanation for the method of bandwidth calculation or selection. In the simulation test, the load spectrum before and after load extrapolation is compared with that under two load spectrums. The error of part life mileage under load spectrums before and after load extrapolation is small, which makes it difficult to explain the reasonableness of load spectrum extrapolation and the validity of load spectrum preparation method. Therefore, there is no certain engineering guidance value.In this paper, the transmission of hybrid electric vehicle (HEV) in E-CVT form, which is widely used at present, is always the research object. Currently, the transmission is highly integrated, with the input shaft of the power source, the power coupling and the main drive integrated into the transmission assembly. For each mixed mode has a specific power flow transmission route, this paper divides the load spectrum into load spectrum blocks of each mixed mode, and then fits, extrapolates and simulates them respectively, fully considering the particularity of different mixed modes. In the extrapolation section, the method of load spectrum extrapolation is explained in detail. Firstly, the optimal bandwidth of kernel density estimation model is calculated based on thumb rule, and then the initial calculation of kernel density is carried out. Then, the adaptive factor is introduced. The size of the factor is related to the density of the points to be estimated. Then, an adaptive bandwidth kernel density estimation model with load mean and amplitude as two-dimensional variables is established to fit extrapolation. Finally, the load spectrum of the transmission assembly is obtained. For simulation test, the transmission assembly is modeled and analyzed, and the simulation results are evaluated based on the actual use of users by accumulated after-sales maintenance statistics for many years. It has higher reliability and authenticity and has certain engineering application guiding value.
Point 2: What is the main novelty of paper? it is better to clearly mentioned in the abstract and introduction.
Response 2: Thank you for your comment, we have supplemented the following text in the summary section: The two-dimensional kernel density estimation model with adaptive bandwidth is used to fit and extrapolate the load-rain-flow matrix of each operation mode of a hybrid electric vehicle, which solves the problem that it is difficult to fit each operation mode of a hybrid electric vehicle due to the complex shape of the rain-flow matrix.
Point 3: There are some different fonts in the text. please correct them based on template.
Response 3: Thank you for your comment, we have rechecked based on the template.
Point 4: What are the findings of the papers? mention them in short in abstract.
Response 4: Thank you for your comment, we add the following words in the abstract: kernel density estimation model proposed in this paper can well fit the rain flow matrix of PHEV load spectrum. The extrapolated load spectrum based on this model has high accuracy and accuracy.
Point 5: Eliminate lumped references and explain them..
Response 5: Thank you for your comment, we have rewritten the second paragraph of the introduction section: Internationally, as an important technical means of energy saving and emission reduction, new energy vehicles can reduce the emission of CO2 and thus promote the international goal of carbon neutralization [3]. In recent years, the scale of the new energy automotive industry has been growing. New energy vehicles, including battery electric vehicle (BEV) and plug-in hybrid electric vehicle (PHEV), have become the second most popular power system in the European market [4], and the automotive market has become more and more active [5]. Compared with pure electric vehicles, plug-in hybrid electric vehicles do not have the problem of "mileage anxiety", while taking into account the advantages of pure electric vehicles when traveling short distances [6]; On the other hand, PHEV still has the fuel-saving performance of traditional hybrid electric vehicles after the energy balance of the rechargeable energy storage system, which attracts wide attention from Chinese enterprises [7], including the national key laboratories of Hubei Province [8] and Chongqing [9], which are famous in China. In 2020, China's PHEV production and sales in the world were 260,000 and 250,000 respectively.
Point 6: It is better to have nomenclature part due to the large numbers of abbreviations.
Response 6: Thank you for your comment, we have added nomenclature part.
Point 7: The motivation part in the introduction is weak. After explaining different prametric and nonparametric rain flow methods, the weak points and disadvantages must be mentioned and then the advantages of proposed method must be highlighted.
Response 7: Thank you for your comment, we add an eighth paragraph to the introduction section, as follows: For the loads of different mixed modes of PHEV, the form of rain flow matrix is quite complex, and the mean and amplitude of loads do not always conform to any distribution. At this time, the parameter rainflow method can not fit the mean and amplitude of these loads very well. Based on the idea of kernel density estimation, instead of hypothetical fitting of the distribution of mean and amplitude of load, the method of non-parametric rain flow extrapolation can estimate the probability density function of input data only by taking the mean and amplitude of load as input variables and selecting the corresponding core function and bandwidth, which is suitable for the loads with strong randomness and complex shape of rain flow matrix. In the field of automotive transmission, especially special transmission assemblies for new energy vehicles, the operation modes are various and complex, and the driving conditions are also variable. The distribution of load averages and amplitudes for some modes of operation does not conform to any existing distribution and can not be assumed in advance. Nonparametric extrapolation can solve this problem very well. Therefore, the non-parameter extrapolation method of rain flow extrapolation is used to extrapolate the load spectrum of PHEV transmission.
Point 8: There are some grammatical and typo errors. Please check the paper using a native spoken/software.
Response 8: Thank you for your comment, we have rechecked and fixed errors with software
Point 9: The last paragraph of introduction must be dedicated to paper structure explanation.
Response 9: Thank you for your comment, we supplement the paper structure explanation part of the paper in the last paragraph of the introduction, as follows: The main contributions of this paper are as follows:
In this paper, the load data measurement test is designed by analyzing the hybrid power system of the test vehicle and the user data provided by the national standard. Dif-ferent operating modes are fitted and extrapolated respectively, and then the extrapolated rainflow matrix is converted into a one-dimensional load spectrum, and simulation fatigue tests are carried out based on the load spectrum. effectiveness.
The structure of this paper is as follows:Section 2 introduces the basic parameters of the test vehicle and the hybrid power system, the method of data preprocessing, the two-dimensional kernel density estimation model, the non-parametric extrapolation method of the load spectrum, and the preparation of the eight-level program load spectrum. In Section 3, the transmission assembly is modeled, and fatigue simulation tests are performed based on the compiled load spectrum. In Section 4, the results of the simulation are discussed according to the characteristics of the operating mode and are validated based on after-sales maintenance statistics. Finally, come to the conclusion
Point 10: Mentioning tested or simulated parameters in Methodology section is not right, it is better to move them to the simulation section.
Response 10: Thank you for your comment, we have added the Load Spectrum Simulation Application section.
Point 11: There is no explanation about Fig.1 inside the text!
Response 11: Thank you for your comment, we interpret Figure 1 as follows: The test vehicle is mainly driven by the engine and drive motor. VCU (Vehicle control unit) controls the engine controller EMS (Engine management system) and motor controller MCU (Motor control unit) to control the work of the engine and drive motor. In addition, the vehicle controller VCU also controls the battery management System BMS (Battery management system), thereby controlling the charge and discharge of the battery. The data exchange of each controller is carried out on the CAN (Controller area network) bus. The combination of the clutch can couple the power output from different power sources, couple the power of the engine with the power of the drive motor, output it to the reducer gear set through the intermediate shaft, and finally drive the car to the drive axle. The combination and separation of the clutch will directly determine the transmission route of the mechanical flow of the transmission assembly.
Point 12: Use the alignment for the rows of paragraphs.
Response 12: Thank you for your comment, revised the alignment for the rows of all paragraphs.
Point 13: In Fig. 2, it is better to put a b for each paper and mention which one is after or before case.
Response 13: Thank you for your comment, added table a and b to explain which one is after or before case.
Point 14: The fonts and the phase of written words in Fig.3 axes are hard to be realized. Please make them more clear.
Response 14: Thank you for your comment, the fonts and axes of Figures 3 and 4 have been reworked
Point 15: Define all parameters in the equations, (for example "Landa" in eq.6)
Response 15: Thank you for your comment,we have checked and defined all parameters in the equation.
Point 16: The font size of Eq. 10 and 11 muse be decreased.
Response 16: Thank you for your comment, we have reduced the font size.
Point 17: The discussion and explanation about obtained results and figures are very short. They need to be improved.
Response 17: Thank you for your comment,we have expended the discussions of results and added after-sales maintenance statistics to supplement,as follow:
Taking the after-sales maintenance data of the car from listing to the present as an auxiliary proof, taking wheel1 as an example, a total of 189 failures have occurred in wheel1 in the transmission assembly from listing to the present, and the average failure mileage is 627958KM. Taking the failure data as a sample, when the confidence level is 99.5%, the confidence interval is calculated as 565162.2KM-690753.8KM. It can be considered that 99.5% of the wheel 1 failure mileage is distributed between 565162.2KM-690753.8KM. Similarly, based on the maintenance statistics of other parts, the confidence interval of other parts with a confidence level of 99.5% can be calculated, as shown in the table7.
Table7. After sales sample number and confidence interval of mileage
|
Gear |
Simulation results /KM |
Number of samples |
Confidence interval /KM |
|
|
wheel1 |
688500 |
189 |
565162.2-690753.8 |
|
|
pinion1 |
689700 |
114 |
574847.1-702590.9 |
|
|
wheel2 |
699600 |
188 |
630884.7-771081.3 |
|
|
pinion2-1 |
681300 |
152 |
573489.9-700932.1 |
|
|
pinion 2-2 |
645600 |
98 |
576282.6-704345.4 |
|
|
wheel 3 |
639900 |
128 |
550152-672408 |
|
|
pinion 3 |
662400 |
177 |
5682743.1-6945574.9 |
|
The life mileage of each part calculated from the simulation results is within the confidence interval, indicating that the faulty parts and their fault mileage life in the simulation test results are basically consistent with the results fed back by the actual users. To sum up, the method for compiling load spectrum of PHEV transmission assembly proposed in this paper has high accuracy and authenticity.
Point 18: Also please clearly mention the main findings of paper even in quantitative mode in conclusion section.
Response 18: Thank you for your comment, we supplement the content in conclusion section as follows: Taking the after-sales statistical maintenance data of this model from listing to now as an auxiliary proof, the failure mileage interval of each gear of the transmission assembly under the confidence of 99.5% is calculated. The simulation results are as follows: the mileage lives of wheel1, pinion1, Wheel2, pinion2-1, pinion 2-2, wheel 3 and pinion 3 are 688500km, 689700 km, 699600 km, 681300 km, 645600 km, 639900 km and 66240 km respectively. The results of the simulation test are all within the confidence interval, indicating that the failure components and the failure mileage life of the simulation test results are basically consistent with the results of the actual user feedback.
In this paper, a method of compiling load spectrum is proposed, which designs the load data test associated with user habits, completes the extrapolation of two-dimensional load spectrum and the compilation of eight level program spectrum. Based on the analysis of simulation results, the durability of transmission assembly can be predicted, forming a complete set of load spectrum compilation process and reliability simulation test application method. To sum up, the two-dimensional kernel density estimation model with adaptive bandwidth can well fit and extrapolate the hybrid modes of hybrid electric vehicles. The fatigue simulation analysis based on the load spectrum compiled by this extrapolation method is basically consistent with the feedback results of actual users. Therefore, the load spectrum of PHEV transmission compiled by using the extrapolation method of two-dimensional kernel density estimation model with adaptive bandwidth has high accuracy. The compilation process of load spectrum proposed in this paper has certain engineering guiding significance.
Round 2
Reviewer 2 Report
There are still some shortcomings and ambiguities as follows:
- In abstract "The extrapolated load spectrum based on this model has high accuracy and accuracy."
- In Figure 2, the simulation time is not same!! why? it should be same for before and after preprocessing.
- Nomenclature part must include not only acronyms but also indexes and symbols.
Author Response
Response to Reviewer 2 Comments
Point 1: In abstract "The extrapolated load spectrum based on this model has high accuracy and accuracy."
Response 1: Thank you for your comment, We have fixed bugs as follows:The extrapolated load spectrum based on this model has high accuracy and authenticity.
Point 2: In Figure 2, the simulation time is not same!! why? it should be same for before and after preprocessing.
Response 2: Thank you for your comment,the preprocessing includes two tasks, one is to remove the invalid amplitude, and the other is to remove the burr. The method of deburring does not change the load and time length. However, the way to remove the invalid amplitude is to directly delete the load less than the threshold, so some load time data points will be deleted, so the time after preprocessing will be shortened.
Point 3: Nomenclature part must include not only acronyms but also indexes and symbols.
Response 3: Thank you for your comment, we have revised the nomenclature part.
